# Lactic Acid Bacteria Fermented *Cordyceps militaris* (GRC-SC11) Suppresses IgE Mediated Mast Cell Activation and Type I Hypersensitive Allergic Murine Model

**DOI:** 10.3390/nu13113849

**Published:** 2021-10-28

**Authors:** Abdul-Rehman Phull, Kyu-Ree Dhong, Hye-Jin Park

**Affiliations:** 1Department of Food Science and Biotechnology, College of BioNano Technology, Gachon University, Seongnam 13120, Gyeonggi-do, Korea; ab.rehman111@yahoo.com; 2Department of Life Science, College of BioNano Technology, Gachon University, Seongnam 13120, Gyeonggi-do, Korea; savanna123@gachon.ac.kr

**Keywords:** *Cordyceps militaris*, anti-allergy, fermentation

## Abstract

*Cordyceps militaris* (*C. militaris*) has various biomedical applications in traditional oriental medicine for different diseases including inflammatory and immune-dysregulated diseases. It is a reservoir of nutritional components such as cordycepin, polysaccharides, and antioxidants. To improve its bioactivity, we fermented *C. militaris* with a *Pediococcus pentosaceus* strain isolated from a salted small octopus (SC11). The current study aimed to evaluate whether *P. pentosaceus* (SC11) fermentation could enhance the anti-allergic potential of *C. militaris* cultured on germinated *Rhynchosia nulubilis* (GRC) against a type I hypersensitive reaction in in vitro and in vivo studies. Total antioxidant capacity and cordycepin content were significantly increased in GRC after SC11 fermentation. GRC-SC11 showed significantly enhanced anti-allergic responses by inhibiting immunoglobulin E (IgE)/antigen-induced degranulation in RBL-2H3 cells, compared to GRC. The results demonstrated the significant inhibition of phosphorylated spleen tyrosine kinase (Syk)/ p38/GRB2-associated binding protein 2 (Gab2)/c-jun in IgE/Ag-triggered RBL-2H3 cells. Furthermore, suppressed mRNA levels of interleukin-4 (IL-4) and tumor necrosis factor-α (TNF-α) in IgE/Ag-activated RBL-2H3 cells were observed. GRC-SC11 significantly ameliorated IgE-induced allergic reactions by suppressing the ear swelling, vascular permeability, and inflammatory cell infiltration in passive cutaneous anaphylaxis (PCA) BALB/c mice. In conclusion, GRC fermented with *P.pentosaceus* exerted enhanced anti-allergic effects, and increased the cordycepin content and antioxidants potential compared to GRC. It can be used as bio-functional food in the prevention and management of type I allergic diseases.

## 1. Introduction

*Cordyceps militaris (C.militaris)* is an important member of the *Clavicipitaceae* family. It is consumed as a traditional medicine for the treatment and management of several diseases, including inflammation and cancer, in East Asia [1,2]. *C. militaris* has various bioactive compounds including cordycepin, polysaccharides, adenosine, and ergosterol [3,4]. Due to exceptional habitats, such as dead insects, it is hard to find naturally grown *C. militaris* in the environment. However, the low extract yield and higher cost are the main hurdles for the commercial utilization of *C. militaris*. 

It has been proven that fermented foods improve their bio-functionalities [5]. *Lactobacillus* strains isolated from dairy food products are used for a variety of food fermentation and applications [6]. Traditional food is considered to have several health benefits, and researchers have been striving to show the augmented biological capabilities of lactic acid bacteria (LAB) fermented foods. These bacterial strains are Gram-positive in their nature and may be found in a variety of foods, including meat, dairy products, and the digestive tract of humans. Various prebiotics, probiotics and postbiotics are known to possess anti-inflammatory, anti-carcinogenic, and other bioactivities [7]. Furthermore, LAB-associated fermentation can improve taste, texture, flavor, and preservation time, as well as physiological activities, including digestion efficiency and natural substance metabolism [8]. *Cordyceps militaris* using germinated *Rhynchosia nulubilis* (GRC) fermented with various microbial strains can be employed without costly extraction techniques and exhibits increased bio-activity. Our group used *R. nulubilis* as a culture substrate for the growth *C. militaris*, instead of the use of dead insects, and subsequently fermented it with *Pediococcus pentosaceus* isolated from the salted small octopus (SC11).

Approximately 20% of the world’s population is affected by one or more allergic conditions. Various allergic conditions, such as anaphylaxis, allergic asthma, allergic rhinitis, atopic dermatitis, and food allergies, are also increasing at an alarming rate [9]. Although the signs of these allergic diseases vary, these conditions share common mechanisms at the molecular level. Mast cells are recognized for the manufacture and secretion of allergy-associated mediators, like histamine, chemokines, cytokines, and growth factors, which play a significant role in allergic diseases [10]. These components are found in vascularized tissues, particularly near surfaces exposed to the external environment, including the skin, gastrointestinal tract, and others [11,12]. The antigen cross-linking of immunoglobulin E (IgE) bound to FcεRI is required for mast cell activation. By cross-linking high-affinity IgE receptors, mast cells undergo degranulation, synthesis, and secretion of allergic related mediators, like histamine, cytokines, chemokines, and several enzymes [13,14]. Activated spleen tyrosine kinase (Syk), an Src family kinase, induces the phosphorylation of phosphoinositide 3-kinase, which induces phospholipase C (PLC) and serine-threonine kinase (Akt), preceded by calcium mobilization and activation of protein kinase C, mitogen-activated protein kinases (MAPKs), and nuclear factor (NF)-κB [15]. Among other mediators, histamine is a key component in acute allergic reactions because it induces vasodilation and enhanced vascular permeability, resulting in edema, hypothermia, and leukocyte recruitment. Mast cells release chemotactic and pro-inflammatory mediators, such as the tumor necrosis factor (TNF)-α, interleukin (IL)-4, IL-1, and IL-8, during the late phase of the allergic phase [16]. Cytokines produced by mast cells alter the terminal microenvironment. Consequently, decreasing pro-inflammatory cytokines is critical for the management and treatment of allergic complications.

Despite the fact that the anti-allergic effects of *C. militaris* have been previously published, no comparative analysis of the anti-allergic effects of GRC-SC11 has been conducted so far. IgE/Ag-mediated passive cutaneous anaphylaxis (PCA) is one of the well-studied paradigms for testing the type-1 hypersensitivity in vivo [17]. Mast cells are also used to study anti-allergic responses linked to the reduced mast cell degranulation and expression of inflammatory cytokines. In this study, we investigated whether *P. pentosaceus* (SC11) fermentation on GRC enhanced its bio-functional nature, and inhibited type I hypersensitivity reactions in RBL-2H3 cells and in mice.

## 2. Materials and Methods

### 2.1. Production of GRC Fermented with P. pentosaceus

The extract of the GRC sample was produced using patented technologies (Cell Activation Research Institution, Seoul, Korea; voucher specimen Kucari: 0903). Fermentation of GRC was carried out as per the previously described method (Kwon et al., 2018) using lactic acid bacterial strain *P. pentosaceus* isolated from the salted small octopus kindly gifted from Prof. Dr. Y.S. Park. The water extract of GRC (5% *w/v*) was prepared at 100 °C for 2 h and then cultured with the SC11 bacterial strain for 24 h. *P. pentosaceus* strains were heat inactivated for 10 min at 100 °C. After being sonicated for 3 min (Sonics & Materials Inc., Newtown, CT, USA), the extract was filtered through 0.45 μM filters to remove bacteria and then was sterilized.

### 2.2. Cell Viability Assay

A cell viability assay was carried out to evaluate whether GRC-SC11 had any toxic effects on RBL-2H3 cells, as in the previously described method [18]. The assay was performed using Cell Counting Kit-8 (CCK-8 (WST-8); Dojindo Laboratories, Kumamoto, Japan). Briefly, cells were plated at a concentration of 2 × 10^4^/well in a 96-well plate and incubated in a carbon dioxide (CO_2_) humidified incubator (Thermo Fisher Scientific 3111, Thermo Fisher Scientific, Waltham, MA, USA) at 37 °C overnight. Viability was determined by measuring the absorbance at 450 nm using a microplate reader (Epoch; Biotek Instruments Inc., Winooski, VT, USA). Non-treated cells (control) were considered 100% viable. The treated cells were compared with the control and expressed as a percentage relative to that of the control.

### 2.3. β-Hexosaminidase Assay

The *β*-hexosaminidase assay was performed as per the previous study [15]. RBL-2H3 cells were treated with 200 ng/ml of dinitrophenyl (DNP) specific IgE (Sigma-Aldrich, St. Louis, MO, USA). After being incubated overnight, cells were washed with the PIPES buffer (25 mM PIPES at pH 7.2, 119 mM NaCl, 5 mM KCl, 1 mM CaCl_2_, 0.4 mM MgCl_2_ 6H_2_O, 40 mM NaOH, 5.6 mM glucose, and 0.1% bovine serum albumin (BSA)). GRC, GRC-SC11, or PP2 (Calbiochem, La Jolla, CA, USA) were treated for 30 min. Cells were stimulated with DNP-BSA (200 ng/ml, antigen; Sigma-Aldrich, St. Louis, MO, USA) for 15 min at 37 °C. The absorbance was measured at 405 nm by a microplate reader (Epoch BioTek Instruments, Winooski, VT, USA).

### 2.4. Reverse Transcription-Polymerase Chain Reaction (RT-PCR)

The RT-PCR was performed by following the previously described method [19,20]. Total RNA from RBL-2H3 cells was prepared using the TRIzol reagent (Invitrogen, Carlsbad, CA, USA) and reverse-transcribed using the ReverTra Ace qPCR RT kit (Toyobo Biologics Inc., Osaka, Japan). A polymerase chain reaction (PCR) was performed according to the manufacturer’s protocol (Qiagen, Hilden, Germany). The PCR program for IL-4 was performed as follows: initial denaturation at 94 °C for 2  min, followed by 30 cycles of denaturation at 94 °C for 20  s, annealing at 56 °C for 10  s, and extension at 72 °C for 25  s, with a final extension at 72 °C for 5  min. The PCR program for TNF-α was performed as follows: “initial denaturation at 94 °C for 2  min, followed by 30 cycles of denaturation at 94 °C for 20 s, annealing at 62.2 °C for 10  s, and extension at 72 °C for 45  s, with a final extension at 72 °C for 5  min. The PCR program for glyceraldehyde 3-phosphate dehydrogenase (GAPDH) was performed as follows: “initial denaturation at 94 °C for 2  min, followed by 30 cycles of denaturation at 94 °C for 20  s, annealing at 62 °C for 10 s, and extension at 72 °C for 25  s, with a final extension at 72 °C for 5  min. The following primers were used: TNF-α forward “(5′-CACCACGCTCTTCTGTCTACTGAAC-3′)”; TNF-α reverse “(5′-CCGGACTCCGTGATGTCTAAGTACT-3′)”; IL-4 forward “(5′-ACCTTGCTGTCACCCTGTTC-3′)”; IL-4 reverse “(5′-TTGTGAGCGTGGACTCATTC-3′; and GAPDH reverse 5′-GACCACAGTCCATGCCATCACTG-3′)” (Cosmo Genetech, Seoul, South Korea). Finally, PCR products were resolved by electrophoresis using 1.5% agarose gels. The resolved bands were examined under the LI-COR Odyssey system (LI-COR Biosciences Inc., Lincoln, NE, USA).

### 2.5. Immunoblotting

Western blotting was performed as described following the previous method [21]. Cells (1 × 10^6^ cells/well) were harvested and homogenized in the radioimmunoprecipitation assay cell lysis buffer (Cell Signaling Technology, Beverly, MA, USA). Protein concentrations were measured using a BCA protein assay kit (Thermo Scientific, Rockford, IL, USA). Proteins (80 μg) were resolved using 10% sodium dodecyl sulfate-polyacrylamide gel. Subsequently, resolved proteins were transferred to nitrocellulose membranes. Membranes were blocked in 5% BSA for one hour. The blots were washed with Tris-buffered saline with Tween 20 buffer (TBST) and incubated with primary antibodies: phosphorylated-Syk (Cell Signaling Technology), phosphorylated-Gab (Cell Signaling Technology), phosphorylated c-Jun (Cell Signaling Technology), and β-actin (Cell Signaling Technology). The immunoblots were washed in TBST(Bio-Rad Laboratories, Hercules, CA, USA) and incubated with horseradish peroxidase-labeled secondary antibody (Cell Signaling Technology). The immuno blots were developed using EZ west Lumi plus (Atto, Tokyo, Japan). Finally, developed immuno blots were visualized, and a densitometry analysis of the bands was performed using LI-COR Odyssey (LI-COR Biosciences, Lincoln, NE, USA).

### 2.6. Experimental Animals

Seven-week-old male BALB/c mice were obtained from a Korean company (Orient Bio Inc., Seongnam, Korea). All the experimental animals were kept in aluminum cages under controlled pathogen-free conditions, such as 55 ± 5 humidity, 12 h light/dark cycle, 24 ± 2 °C temperature, and provided with standard feed along with water ad libitum. Animals were maintained in accordance with the guidelines of the Institutional Animal Care and Use Committee of Gachon University, Seongnam, Gyeonggi-do 461-701, South Korea (GIACUC- R2018001). All experimental mice were separated into four groups and comprised of six male BALB/c mice. Specified doses of GRC-SC11 and the reference drug were administered orally by gavage. 

Group I: Normal control (non-treated)

Group II: Passive cutaneous anaphylaxis (PCA) disease control (0.5 µg IgE, anti- DNP IgE sensitized and challenged with 200 µg DNP-HAS + 2% Evans blue solution)

Group III: PCA mice treated with GRC-SC11 (100 mg/kg) 

Group IV: PCA mice treated with the reference drug control (10 mg/kg).

#### 2.6.1. PCA BALB/c Mice Model

The PCA assay was performed as described previously in a study [22]. For the PCA reaction, BALB/c mice were intravenously injected with PBS solution containing 200 μg DNP-BSA and 2% (*w/v*) Evans blue solution 24 h after the intradermal injection of 0.5 μg anti-DNP IgE into one ear per mouse. In order, to investigate the inhibitory potential in the animals, each PCA mouse was orally administered with 100 mg/kg of sample one hour before DNP-BSA. Cetirizine was used as a standard drug at the concentration of 10 mg/kg. One hour after the challenge, the mice were euthanized using CO_2_ and sacrificed. The ears were collected for histological investigation and extraction of the Evans blue dye. Extravasated Evans blue dye was extracted by incubating the ear biopsies in formamide at 60 °C for 16 h. Optical densities were recorded at 620 nm using a microplate reader (Tecan, Männedorf, Switzerland).

#### 2.6.2. Histopathologic Assessment

Histopathological investigations were carried out using a previously described method with slight modifications [22]. The ear biopsies were fixed in formalin (10%), embedded in paraffin, and stained with hematoxylin and eosin (Sigma-Aldrich, St. Louis, MO, USA). Generally, the extracellular matrix and cytoplasm were stained pink with eosin, while the nuclei were stained blue with the hematoxylin dye. The images of hematoxylin and eosin-stained tissues were taken using a Nikon Eclipse Ti incorporated CCD camera at 100× and 200× (Nikon, Melville, NY; Point Grey Research Inc., Richmond, BC, Canada). Stained cells were measured using the ImageJ software (1.46, National Institutes of Health, Bethesda, MD, USA).

### 2.7. High-Performance Liquid Chromatography (HPLC) Analysis for Cordycepin

Cordycepin (3’-deoxyadenosine) was determined in GRC and GRC-SC11 using HPLC (Agilent 1100 liquid chromatography system: Palo Alto, CA, USA). The measurement data were provided by the Korea Basic Science Institute (KBSI) on the Ochang Campus, South Korea. Waters Acquity BEH C18 columns (100 mm × 2.1 mm id, 1.7 µm and 12.5 mm × 4.6 mm id, 5 µm) were used. Water and methanol were used as mobile phases with a flow rate of 1.0 mL/min. UV detection was performed at the 254 nm wavelength. A calibration curve was prepared by plotting the peak area against the concentration using different concentrations of cordycepin (Sigma-Aldrich, St. Louis, MO, USA). 

### 2.8. Total Antioxidant Capacity (TAC) Assay

The TAC of the samples was investigated using the phosphomolybdenum method. the TAC activity was expressed as ascorbic acid equivalents (AAE). Briefly, 100 µL of the extract and 400 µL of the reagent solution (ammonium molybdate (4 mM), sodium phosphate (28 mM), and sulfuric acid (0.6 M)) were mixed. The assay mixture was incubated at 95 °C for 90 min and cooled at an ambient temperature, and the optical density was recorded at 695 nm using an OPTIMax tunable microplate reader (Molecular Devices; Sunnyvale, CA, USA). Ascorbic acid was used as the reference antioxidant, and ascorbic acid equivalents (AAE) were calculated using a standard curve. The antioxidant activity of the sample was expressed as μg or mg ascorbic acid equivalent per mg of extract (μg AAE/mg or mg AAE/g).

### 2.9. Statistical Analysis

Results are presented as representative or mean ± standard deviation (SD) and were analyzed by using a one-way analysis of variance (ANOVA), followed by Duncan’s *t*-test, Student’s *t*-test, or Dunnett’s *t*-test. The normality distribution of all data was checked by the Shapiro Wilk test. Data were analyzed using the SPSS v.12 software (SPSS Inc., Chicago, IL, USA). 

## 3. Results

### 3.1. Effect of P. pentosaceus SC11 Mediated Fermentation on the Antioxidant Potential and Cordycepin Content of Cultured C. militaris

Fermentation has various beneficial effects in improving the nutritional and bio functional properties of food. To examine the antioxidant activities of GRC and GRC-SC11, the extracts of both samples were examined using a TAC assay. Fermentation improved the antioxidant potential of GRC-SC11, which exhibited higher activity than GRC (Figure 1). In addition, increased cordycepin contents as 126.46 ± 6.52 µM and 179.49 ± 10.68 µM were observed in GRC-SC11, respectively. These results are shown in Figure 1.

### 3.2. Effect of GRC and GRC-SC11 on IgE/Ag-Activated Degranulation of RBL-2H3 Cells

To select the best LAB strain for fermenting GRC, we compared the inhibitory activity on degranulation of IgE/Ag-stimulated RBL-2H3 cells. Among them, P. pentosaceus SC11 exerted the highest inhibitory activity on the release of β-hexosaminidase (Appendix A); therefore, we chose P. pentosaceus SC11 for fermenting GRC for further anti-allergic study. To evaluate the anti-allergic potential of GRC and GRC-SC11 extracts, we examined their effects on IgE/Ag-triggered degranulation in RBL-2H3 cells. β-hexosaminidase release was quantified [19]. The results showed that GRC-SC11 exhibited a higher inhibitory effect on β-hexosaminidase release through the degranulation index (Figure 2A). To investigate the effect of GRC-SC11 on cell viability, a CCK8 assay was performed. RBL-2H3 cells were untreated or treated with different concentrations of GRC-SC11 for 24 h. The results showed that GRC-SC11 did not exhibit any toxic effect and subsequently did not alter the viability of RBL-2H3 cells up to 300 μg/mL (*p* < 0.05) (Figure 2B). Henceforth, GRC-SC11 was selected for further study.

### 3.3. Effect of GRC-SC11 Pro-Inflammatory Cytokines in IgE/Ag-Stimulated RBL-2H3 Cells

We assessed GRC-SC11 reduced IgE/Ag-stimulated TNF-α and IL-4 mRNA expression through RT-PCR in RBL-2H3 cells. The IgE/Ag-stimulated production of IL-4 and TNF-α in RBL-2H3 is associated with the synthesis of inflammatory mediators, such as reactive oxygen species, nitric oxide, chemokines and other cytokines, and induces IgE antibody generation in B cells [23]. GRC-SC11 significantly suppressed TNF-α mRNA expression in a dose-dependent manner. GRC-SC11 also decreased IL-4 mRNA expression in RBL-2H3 treated with IgE/Ag. (Figure 3A,B). 

### 3.4. Effect of GRC-SC11 on IgE/Ag Associated Signaling Molecules in RBL-2H3 Cells

The crosslinking of Ag with IgE-stimulated FcεRI receptors in mast cells leads to the triggering of diverse intracellular signaling cascades, inducing the degranulation and production of pro-inflammatory cytokines. During the FcεRI signaling pathway, activated Src family kinases such as Lyn and Syk phosphorylate downstream molecules (GRB2-associated binding protein 2 (IL-4)) and mitogen-activated protein kinase (MAPK) [24]. To elucidate the anti-allergic effect of GRC-SC11 on the activation of the FcεRI and MAPK signaling pathways, we assessed whether GRC-SC11 decreased the levels of phosphorylated Syk, Gab2, p38, and c-Jun proteins in RBL-2H3 cells treated with IgE/Ag. GRC-SC11 significantly suppressed the levels of phosphorylated Syk, Gab2, p38, and c-Jun proteins, compared to IgE/Ag-stimulated controls (Figure 4A,B, *p* < 0.001 and *p* < 0.05). The findings of this study indicated that GRC-SC11 downregulated allergy associated signaling molecules of FcεRI and MAPK pathways in IgE/Ag-stimulated RBL-2H3 cells.

### 3.5. Effect of GRC-SC11 on the PCA Model

A PCA mouse model was used to study the immediate-type allergic reactions (Figure 5A). To induce PCA in the murine ear, mice were stimulated with IgE in the ear and this was challenged by injecting DNP-BSA containing 1% Evans blue dye into the tail vein. Blue spots were developed in the ear of mice stimulated with IgE/Ag. This was due to the augmented vascular permeability induced by histamine released from RBL-2H3 cells. The results demonstrated a prominent increase in Evans blue staining in the PCA model (*p* < 0.001). GRC-SC11(100 mg/kg) and the reference drug (10 mg/kg) significantly suppressed dye extravasation (*p* < 0.001) (Figure 5B,C). The data showed that GRC-SC11 decreased the IgE/Ag-stimulated PCA response in BALB/c mice.

### 3.6. Effect of GRC-SC11 on Histopathological Changes and Ear Swelling Response in the PCA BALB/c Mice Model

To examine the consequences of GRC-SC11 on ear tissue and ear swelling response in a PCA animal model, the ear biopsies were stained with hematoxylin and eosin. Skin swelling due to augmented vascular permeability-associated histamine secretion from cells at the PCA site was observed, and this effect could also be attributed to the increased number of infiltrated immune cells. GRC-SC11(100 mg/kg) and the reference (10 mg/kg) significantly reduced the ear thickness and infiltrated immune cells compared to that in the PCA mice model. Collectively, these data showed that GRC-SC11 attenuated the IgE/Ag-stimulated infiltration of inflammatory cells along with increased dermal and epidermal ear thickness in IgE/Ag-stimulated mice (Figure 6A,B). 

## 4. Discussion

In several developing countries, the prevalence of allergies, such as type I hypersensitivity disorders, is increasing at a higher rate. Genetic and environmental factors are major contributing factors in the development of allergic disorders. Nearly one in every four children in high-income nations have been diagnosed with allergic rhinitis, asthma, or eczema in the last three to four decades [25]. Small molecules, such as antagonists of leukotrienes or histamine receptors as well as steroid therapy, are the most widely used treatments for allergic disorders these days [26]. Traditional treatments, on the other hand, can cause unpleasant side effects, such as drowsiness, dry mouth, and sore stomach. *C. militaris*, a traditional East Asian medicine, is well known for its anti-inflammatory, anti-anemia, and anti-cancer properties. Anti-allergic remedies made from herbal constituents or extracts derived from ancient medicinal plants or fungi are gaining popularity because of their lower toxic effects. *C. militaris*, also known as Dong-Choong-Ha-Cho in Korea, is a medicinal fungus that has long been used to treat cancer [1], hyperlipidemia [27], hepatic cirrhosis [28], bronchitis [29], and asthma [30] in East Asia. 

Food fermentation is an old food processing method that has long been used to increase the shelf life, organoleptic qualities, nutritional content, and bio-functionalities of food items. LAB are capable of producing a large quantity of secondary metabolites with good health advantages. In fact, some microbes can raise the amounts of a variety of beneficial chemicals, including antioxidants, phenolic compounds, peptides, and vitamins. Furthermore, fermented foods contribute to the modulation of the host’s physiological balance and gut microbiota along with bioactive molecules. It is well known that edible fungi and their functional components have health benefits linked to the host gut microbiota. Bioactive antioxidants including glycolipids, aromatic phenols, and fatty acid derivatives may be found in edible fungi [31]. Well-known medicinal fungi, such as *C. militaris*, have been reported to have similar effects against many ailments. Because of its habitat, it is difficult to grow naturally. Recent studies have shown that cultured *C. militaris* contains high levels of amino acids [32]. There are few reports on fermented fungi [32]. Therefore, *C. militaris* grown on *R. nulubilis* was used in this study. Furthermore, recent advancements in the fermentation industry have focused on potentially bioactive chemicals that may be utilized as ingredients in functional food compositions for the treatment and management of various illnesses [33]. This study aimed to improve the bioactive nature of GRC by fermenting with the lactic acid bacterial strain (SC11) and to inhibit type I hypersensitive reactions using both in vivo and in vitro models. The augmented anti-allergic effects of GRC-SC11 might be attributed to its increased cordycepin content, antioxidant potential, and other ingredients, such as phenolic compounds. Herein, the *Pediococcus pentosaceus* bacterial strain was tested for fermenting the cultured GRC. Interestingly, we found that *Pediococcus pentosaceus* could ferment cultured *C. militaris* (GRC) but not naturally occurring *C. militaris* (data not shown). In this study, we developed a method for culturing GRC and GRC-SC11 to test their anti-allergic activities.

Many studies have found that oxidative stress, through the production of oxidant factors and the decrease of antioxidant factors, plays a key role in the development of allergic disorders [34]. Subsequently, altered antioxidant systems and an imbalance in the reactive oxygen species (ROS) are associated with allergy-associated ailments, such as asthma [35]. Activated inflammatory cells, such as eosinophils, neutrophils, monocytes, and macrophages, can generate ROS, which are linked to some extent with hypersensitivity [36]. Reduced antioxidant enzymes, such as superoxide dismutase, catalase, glutathione peroxidase, TAC, and glutathione levels, were found to be substantially lower in the blood and nasal mucosa of allergic rhinitis mice (Li et al., 2011). However, given the potential for traditional antioxidants to have poor pharmacokinetic characteristics, some researchers have proposed the production of novel antioxidant-based treatments for allergic diseases [37,38]. It has also been shown that *C. militaris* reduces the immune-inflammatory response and oxidative stress in IgE/Ag-stimulated allergic responses both in vitro and in vivo. 

One of the previous studies showed that the antioxidant activity of myrtle homogenate fermented with *L. plantarum* C2 was significantly higher than that of the non-fermented homogenate, as observed through the radical scavenging potential. Recently, our group also reported that fermentation increased the antioxidant activities of *R. nulubilis* using *L. pentosus* SC65 and *P. pentosaceus* ON89A bacterial strains [21]. Previous studies have demonstrated that polyphenol content, which is known to have antioxidant and anti-inflammatory properties, was increased by phenolic esterase activity in LAB [39,40]. Our results showed that the content of cordycepin, which is an antioxidant, was higher in GRC-SC11 than in GRC. This was supported by the TAC assay results, SC11 fermentation resulted in an increase in the antioxidants potential, which was >4-times than that found in GRC. In this study, significantly augmented TAC activity and cordycepin content suggest that the anti-allergic effects of GRC-SC11 are, at least partially, mediated by its antioxidant activity. Cordycepin has been identified to be the key constituent of *C. militaris*, which exerts anti-inflammatory activity [41]. Previous studies have reported the protective effects of cordycepin on ovalbumin-induced allergic inflammation by suppressing Th17 responses in ovalbumin-sensitized mice [42]. The enhanced anti-allergic properties of GRC-SC11 are most likely due to antioxidants, such as novel isoflavonoids, cordycepin, and other bioactive constituents.

Mast cells are stimulated by IgE-regulated antigens via IgE specific receptors. Subsequently, the resulting cross linking of receptors, Syk, Gab2, and other downstream tyrosine kinases are activated [24]. As a result, Syk and Gab2 are promising targets for the development of allergy medicines. Activated Syk phosphorylates p38 and c-Jun, which are downstream signaling molecules. The Syk pathway is associated with the generation of pro-inflammatory cytokines [43]. Pro-inflammatory cytokines, such as TNF-α and IL-4 are produced from the activated mast cells [44]. Mast cell activation, leukocyte infiltration, allergic inflammation, and allergy-related processes are all triggered by these pro-inflammatory cytokines. TNF-α is involved in the regulation of allergic inflammation. Meanwhile, IL-4 is required for Th2 response expansion and induced B cell switching to IgE synthesis [45]. Our results showed that GRC-SC11 significantly decreased the protein expression of p-Syk, p-Gab, p-p38, and p-c-Jun in IgE/Ag-stimulated RBL-2H3 cells. Additionaly, GRC-SC11 significantly inhibited pro-inflammatory molecules, such as IL-4 and TNF-α, in IgE /Ag-stimulated RBL-2H3 cells.

Previously, we reported novel isoflavonoid compounds that are similar to genistein or daidzein from *C. militaris* cultured on germinated soybeans (genistein 4-O-β-D-glucoside 4″-O-methylate, daidzein 7-O-β-D-glucoside 4″-O-methylate, genistein 7-O-β-D-glucoside 4″-O-methylate, and glycitein 7-O-β-D-glucoside 4″-O-methylate) [46]. Remarkably, the novel isoflavonoids genistein and daidzein from *C. militaris* were grown on germinated soybeans and prominantly suppressed the degranulation in antigen-activated RBL-2H3 cells [19]. Previously, we observed that genistein 4-O-β-d-glucoside 4″-O-methylate decreased the release of pro-inflammatory cytokines, p-Syk, and phospho-extracellular signal-regulated kinase (p-ERK), in antigen-induced RBL-2H3 cells and suppressed the PCA reaction [47]. It is conceivable that isoflavonoid compounds in GRC-SC11 might inhibit allergic activity by suppressing Syk activation. 

PCA is a type I local allergy stimulated in mice by intradermal inoculating IgE into the ears and intravenous injecting antigen into the tails [22]. The interaction of the antigen and antibody enhanced the synthesis of allergy-associsted chemicals, like histamine in subcutaneous immune cells, as well as boosted the Evans blue dye’s permeability and extravasation that was injected into the bloodstream with the antigen [22]. In comparison to the IgE/Ag-mediated allergy mouse model, we found that GRC-SC11 decreased the amount of extravasated Evans blue dye in the ear (Figure 5). Allergy-related trafficking molecules and endothelial selectins that can bind to immune cells are expressed as a result of IgE-mediated PCA reactions [48] and subsequently immune cells infiltrate the ear tissues [49]. Our results showed that GRC-SC11 substantially reduced the IgE-regulated PCA response by attenuating ear edema and decreasing the quantity of infiltrating inflammatory cells (Figure 6). These findings suggest that GRC-SC11 reduces the vascular expansibility and activates immune cells in the dermis induced by degranulation and produces cytokines of IgE/Ag-triggered mast and basophils cells, which is consistent with our in vitro results. In a previous study, we reported that the novel bioflavonoid genistein, 4-O-β-D-glucoside 4″-O-methylate, from *C. militaris* inhibited proinflammatory cytokines in antigen-triggered RBL-2H3 cells along with the PCA reaction [47].

## 5. Conclusions

We explored how the fermentation of cultured *C. militaris* is a beneficial approach to increase its bio-functionality. The fermentation of cultured *C. militaris* was carried out with the lactic acid bacterium, *P. pentosaceus*, isolated from a salted small octopus. An increase in the TAC of GRC-SC11 (>4 times) and cordycepin content (42%) was observed in GRC-SC11 compared to GRC. We also observed the augmented anti-allergic activity of GRC-SC11 in the IgE/Ag-regulated allergic response by inhibiting the degranulation and decrease in the expression of proinflammatory cytokines (IL-4 and TNF-α mRNA) in RBL-2H3 cells, compared to GRC. Moreover, GRC-SC11 inhibited the phosphorylation of Gab2, p-Syk, and NF-κB in the FcεRI-mediated signaling pathway along with MAPKs (p38/ c-Jun) in the IgE/antigen-stimulated RBL-2H3 cells, compared to GRC. Furthermore, GRC-SC11 demonstrated an anti-allergic activity in the PCA mice model by decreasing the allergy-related infiltration of cells and swelling. Conclusively, SC11-mediated fermentation increased the bio-functional ingredients of GRC, including cordycepin and antioxidants. GRC-SC11 inhibited the degranulation and suppressed the release of pro-inflammatory cytokines in IgE/Ag-activated RBL-2H3 cells. In addition, GRC-SC11 inhibited the signal transduction cascade responsible for type I hypersensitive allergic responses. This is supported by the in vivo experiment in which GRC-SC11 attenuated the IgE-mediated type I hypersensitivity reactions in the ear of BALB/c mice. This study highlights the potential of GRC-SC11 for developing fermented bio-functional foods and health supplements for the treatment and management of type I allergic diseases.

## Figures and Tables

**Figure 1 nutrients-13-03849-f001:**
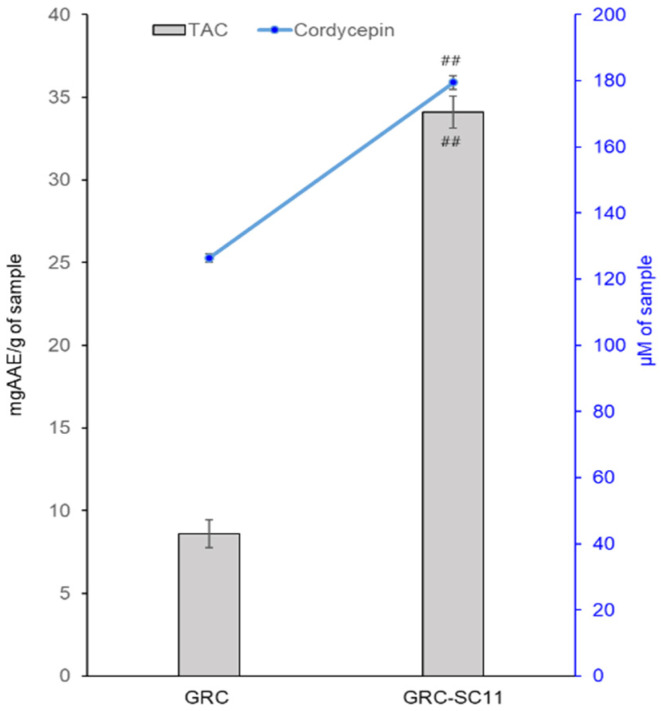
Total antioxidant capacity (TAC) and cordycepin content of *C. militaris* grown on germinated *R. nulubilis* (GRC) and GRC fermented by *P. pentosaceus* isolated from the salted small octopus (SC11) (GRC-SC11). The experiments were repeated three times and results are presented as mean ± SD. TAC was expressed as mg ascorbic acid equivalent (mgAAE). Results were considered significant at the *p* value of ## *p* < 0.01. Data comparisons between the two groups were analyzed using the Student’s *t*-test.

**Figure 2 nutrients-13-03849-f002:**
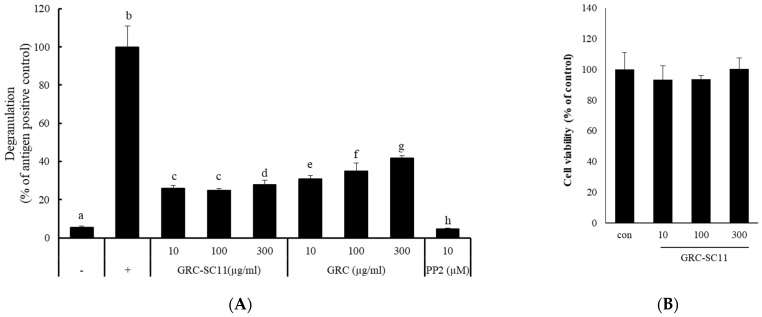
Effects of GRC and GRC-SC11 on degranulation and cell viability of RBL-2H3 cells. (**A**) The release of β-hexosaminidase was calculated from the degranulation of RBL-2H3 cells. RBL-2H3 cells were treated with the indicated concentrations of GRC and GRC-SC11 and 10 µM 4-amino-5-(4-chlorophenyl)-7-(t-butyl) pyrazolo (3, 4-d) pyrimidine (PP2) for 30 min. Control (−) is the untreated control, while control (+) is stimulated with immunoglobulin E (IgE)/Ag. PP2 is an Src family kinase inhibitor that was used as the positive control. Different letters indicate the significant differences between groups. Results were compared for statistical significance through a one-way analysis of variance (ANOVA) followed by Duncan’s *t*-test (*p* < 0.05). (**B**) RBL-2H3 cells were exposed to different concentrations (10, 100, and 300 µg/mL) of GRC-SC11 for 24 h and cell viability was investigated. The experiments were repeated three times and results are presented as the representative or mean ± SD. Results were compared for statistical significance through a one-way analysis of variance (ANOVA followed by Duncan’s *t*-test (*p* < 0.05)).

**Figure 3 nutrients-13-03849-f003:**
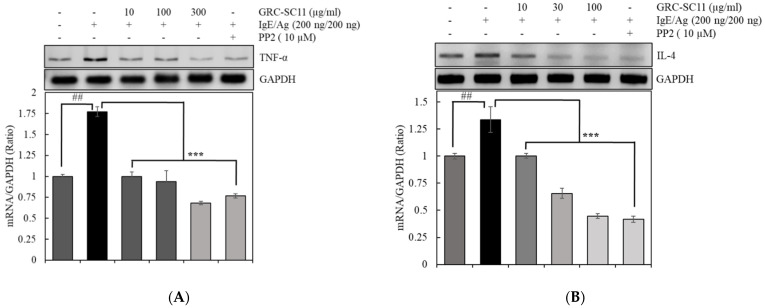
Effect of GRC-SC11 on the production of pro-inflammatory cytokines in IgE/Ag-stimulated RBL-2H3 mast cells. The RBL-2H3 cells were untreated or treated with indicated concentrations of GRC-SC11 in the absence or presence of the 200 ng/mL of IgE/Ag or 10 µM of PP2. GRC-SC11 reduced the expressions of (**A**) tumor necrosis factor (TNF)- α and (**B**) interleukin (IL)-4 in IgE/Ag-stimulated RBL-2H3 mast cells. The expression study was carried out by Western blotting method using GAPDH as the loading control. The relative amounts of TNF- α, IL-4 were analyzed by using the Image J software. The results are presented as the representative or mean ± SD of three independent experiments. Results were analyzed for statistical significance through a one-way ANOVA followed by Dunnett’s *t*-test. The probability value of ## *p* < 0.01 was considered significant compared to the non-treated cells and *** *p* < 0.01 compared to the IgE/Ag-treated cells.

**Figure 4 nutrients-13-03849-f004:**
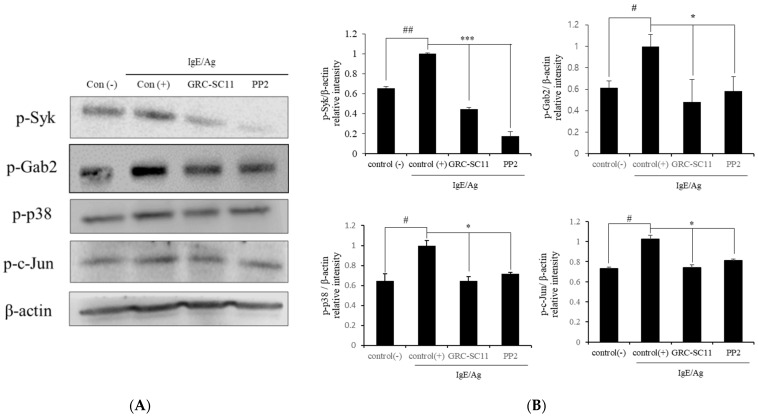
Effects of GRC-SC11 on the signaling molecules IgE/Ag-activated RBL-2H3 cells. RBL-2H3 cells were sensitized with dinitrophenol (DNP)-IgE and then stimulated with antigen for 15 min. (**A**) Western blotting assay of phospho-spleen tyrosine kinase (p-Syk), Syk, phospho-GRB2-associated binding protein 2 (p-Gab2), Gab2, p-p38, and p-c-Jun. (**B**) Expression levels of GRC-SC11. The levels were normalized to β-actin levels. The experiments were repeated three times and results are presented as the representative or mean ± SD. The probability values of # *p* < 0.05 and ## *p* < 0.01 were considered to be significant compared to the non-treated cells, and * *p* < 0.05 and *** *p* < 0.001 compared to the IgE/Ag treated cells. One-way ANOVA was used for the comparison of group means, followed by Dunnett’s *t*-test.

**Figure 5 nutrients-13-03849-f005:**
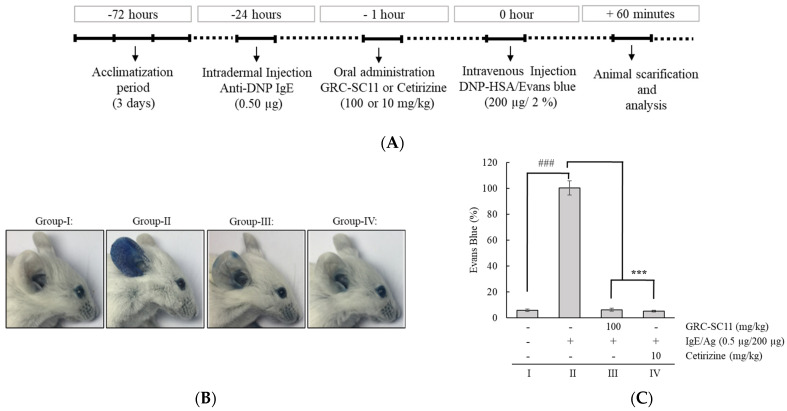
Effect of GRC-SC11 on the IgE/Ag-stimulated passive cutaneous anaphylaxis (PCA) BALB/c mice model. (**A**) Scheme of the PCA design. PCA was induced by subcutaneous injection of 0.5 μg of DNP-specific IgE into the ear of male BALB/c mice for 24 h and followed by the intravenous administration of 200 μg of DNP-bovine serum albumin (BSA) antigen having 2% Evans blue dye in the tail region. Anti-allergic activity of GRC-SC11 was assessed at the concentration of 100 mg/kg, while cetirizine was used as the standard drug (10 mg/kg). Group I: Normal control (non-treated). Group II: PCA disease control (0.5 µg IgE, anti-dinitrophenyl (DNP) IgE sensitized and challenged with 200 µg DNP-HAS + 2% Evans blue solution). Group III: PCA mice treated with GRC-SC11 (100 mg/kg). Group IV: PCA mice treated with the reference drug control (10 mg/kg). (**B**) Representative image of murine ear (*n* = 7 mice/group). (**C**) The mean±SD of extravasated dye (*n* = 7 mice/group). Differences were significant at the level of ### *p* < 0.001 (compared with normal animals) and *** *p* < 0.001 (compared with normal PCA animals). Statistical analysis of results was performed through one-way ANOVA followed by Dunnett’s *t*-test.

**Figure 6 nutrients-13-03849-f006:**
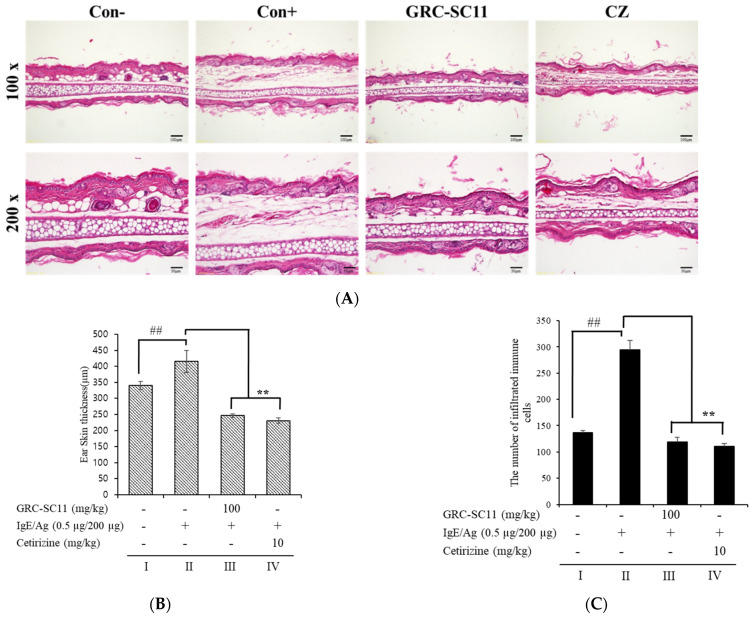
Effect of GRC-SC11 on IgE/Ag-induced ear swelling response in the BALB/c mice models. Histopathological examination of ear tissues of the IgE/Ag-triggered PCA BALB/c mice models. (**A**) Representative images of ear tissues stained with hematoxylin and eosin (*n* = 7/group; magnifications, 100× and 200×) of group-I: normal control, group-II: PCA disease control (IgE/Ag, anti-DNP IgE sensitized and DNP-HAS challenged), group-III: GRC-SC11 (100 mg/kg), and group-IV: drug control (10 mg/Kg). (**B**) A dial thickness gauge was used to measure ear thickness. Data are expressed as mean ± SD (*n* = 7/group). (**C**) Infiltrated immune cells were counted in the ear of mice. Data are expressed as mean ± SD (*n* = 7/group). Results were considered significant at the level of ## *p* < 0.01 (compared with normal animals) and ** *p* < 0.01 (compared with normal PCA model control). Statistical analysis of results was performed through one-way ANOVA followed by Dunnett’s *t*-test.

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
