# Peer review of "Lactic Acid Bacteria Fermented *Cordyceps militaris* (GRC-SC11) Suppresses IgE Mediated Mast Cell Activation and Type I Hypersensitive Allergic Murine Model"

_nutrients, 2021, doi:10.3390/nu13113849_

Round 1

Reviewer 1 Report

General comments: The study focuses in an area that could attain an important or relevant role ameliorating physiopathological signs of allergies. However, the study seems to be in a very early stage baing difficult to demostrate the true nature of the bioative compound responsible for the effects. The discussion section is presented as a summary of the scientific literature. Besides, the conection between the processes studied results inferential. Thus, it can be improved for a better understanding of the potential influence of the results.

Specific comments: Line 14: ‘…Pediococcus pentosaceus probiotic strain…’ The term probiotic implies diverse ecophysiological conditions.

Line 25: ‘…PCA…’. Please, fully describe the abbreviation the first time appeared

Line 26: ‘…may be attributed…’. Conclusion too ambiguous; can or cannot?. Quantify this compound in the supernatants of eliminate it from the sample or report its bioavailability. Results don’t support this conclusion.

Line 33: ‘…ailments..’. Please rewrite.

Line 41: ‘…probiotic…’. This term implies, for example, that bacteria exert beneficial physiological effects, survive to gastric conditions and bile acids as well as colonize, etc. From my point of view it cannot be used just for metabolic products derived from bacterial metabolism. Please, revise the use of the term throughout the manuscript.

Line 155: ‘…PCA…’. Please, describe what this abbreviation means.

Line 157: What is the basis to use this dose? Why authors di not use the same as for the reference drug?

Fig. 1: Please, plot the SD or SEM for cordycepin. Apparently, fermentation only increases ±2 mg/g the presence of this compound. Does it result statistically significant? Otherwise, it must be demonstrated that only this small variation is responsible for beneficial effects.

Line 227: Viability assay is based on cell counting. For this type of approach, it could be more relevant to evaluate metabolic activities.

Fig. 2. Why GRC concentrations are not expressed as µM for cordycepin? Authors attribute the beneficial effects to this compound, but don’t focus on it.

Line 272: ‘…GRC-SC11 showed anti-allergic activity…’. Please, rewrite. This assay shows positive effects ameliorating IgE/Ag-mediated activation of proteins and allergy implies a significant bulk of many other reactions.

Line 299: It is not easy to understand the mean values in panel c as panel b shows clear significantly different blue areas. Why?

Line 338-339: When using sentences such as ‘…has been widely used to treat asthma, 338 bronchitis, hyperlipidemia, cancer, and hepatic cirrhosis in East Asia…’, please, include references.

Line 345: Fermented foods can also be produced by microorganisms, which are not considered probiotics. In fact, not all bifidobacterial and lactobacilli used to produce yogurts are considered probiotics. Please, revise the use of this term throughout the manuscript.

Line 357: ‘…might be attributed…’. Authors only proved that cordyceptin is present, but current data do not support any potential beneficial effects derived from this compound.

Line 364-377: ROS, etc can play a role in allergic processes as these compounds are a hallmark of damaged tissues and cells. However, this does not mean that blocking ROS helps controlling allergies further than preserving cell functionality.

Line 393-395: Authors accept that other compounds than cordyceptin could be responsible of the beneficial effects. The current study is on a early stage to confirm the bioactive compounds.

Line 439: This section is presented as a summary of the results, but What are main conclusions?

Author Response

Thank you for reviewer's comments.

Thank you

Reviewer 2 Report

In their manuscript Phull et al. describe the use of Cordyceps militaris grown on Rhynchosia nulubilis fermented by a probiotic bacterium (Pediococus  pentosaceus) as a mediator to inhibit a Type I allergic response. They use in vitro as well as in vivo systems to show this activity. In principle, inhibitors of allergic responses are very desirable. However, many points need to be addressed.

  1. The manuscript is very difficult to read. This is due to some English problems but mainly due to the style. The authors try to explain for instance the signaling induced by the FcE receptor but the description is very confuse. Similarly, the authors claim that antigen presenting cells activate B cells to generate IgE. What does that mean.

 Abbreviations are used very unsystematically. For instance, GRC is used in the Abstract but explained much later. This holds true for other examples at many locations. For instance, PCA is explained in the discussion. Cordyceptin is a name the does not explain its chemical nature.

  1. It appears odd to use bacteria from salted small octopus. Why was it used and what is the advantage.
  2. Description of material and methods is very unsatisfactorial. How is it possible to crry out an extraction by using 105°C. It is not mentioned that RBL-2H3 is a basophil cell which is taken here as mast cell surrogate. For sure the authors have not used a CO2 incubator but a normal incubator whith ?? CO2. The authors do not mention from where the DNP-BSA was derived from and what the conjugation density was. What was used as DNP specific IgE. The Release assay is not described at all and why was hexosamineidase release used as read out. Was indeed not phosphatase inhibitor used in the Western blot experiments? The principle of the PCA assay is not described. What does Cetirizine do? Suddenly, histology is described in great details.
  3. If I understand correctly, the CRC was fermented with SC11 and then extracted and sonicated. How can the authors rule out that the additional effect is not entirely due to bacterial components? I could not find an appropriate control for that
  4. Figure 2 a. The way the statistics is presented here is completely meaningless.
  5. Line 246-247. Values look correlated with IL-4 and TNF-a and not with the extract.
  6. Figure 5 b. Indicate what treatment the individual groups represent.
  7. In the discussion the authos explain the effects with the activity of cordycepin but later thexy explain the effect with other compounds that are found in germinated soybeans. This is very confusing.

Author Response

Thank you for reviewer's commnents.

Please see the attach ment

Thank you

Round 2

Reviewer 1 Report

Dear authors,

  my comments are highlighted in blue.

Author Response

Thank you for reviewer's comments.

Thank you.

Reviewer 2 Report

The manuscript has very much improved. A few very minor point should be addressed:

  1. Use of a bacterium form salted small octopus: may be in M&M 2.1 should be added: ....which was found superior to other members of this species.
  2. DNP-BSA may be: DNP30-40-BSA.
  3. Figure 2a: the presentation of the statisctis here is still meaningless. In the other Figures it is done the standard way. why not here?
  4. The authors should again very carefully go through the legends of the figures and correct errors. For instance, legend of Figure 5 does not contain a (c) that is found in the Figure. Or Figure 6a showns now very clearly A-H which is not explained in the legend. For Figure 6 (c) a representative (A) and a (B) mean is mentioined. What does that indicate and were is it found in the figure.

Author Response

(The authors gave the same response as above.)
